# Treatment of Heart Failure Patients with Anxiolytics Is Associated with Adverse Outcomes, with and without Depression

**DOI:** 10.3390/jcm9123967

**Published:** 2020-12-07

**Authors:** Donna R. Zwas, Andre Keren, Offer Amir, Israel Gotsman

**Affiliations:** 1Heart Institute, Hadassah University Hospital, Jerusalem 91120, Israel; kerena@mail.huji.ac.il (A.K.); oamir@hadassah.org.il (O.A.); igotsman@bezeqint.net (I.G.); 2Heart Failure Center, Clalit Health Services, Tel Aviv 16250, Israel

**Keywords:** anxiety, depression, heart failure, outcome

## Abstract

Background: Few studies have evaluated the effect of pharmacologic treatment of anxiety on outcomes in heart failure (HF) patients. This study examined the impact of treatment with anxiolytics on clinical outcomes in a real-world sample of HF patients with and without depression. Methods: Patients diagnosed with HF were retrieved from a large HMO database. Patients prescribed anxiolytic medication and patients diagnosed with depression and/or prescribed anti-depressant medication were followed for cardiac-related hospitalizations and death. Results: The study cohort included 6293 HF patients. Treatment with anxiolytics was associated with decreased one-year survival compared to untreated individuals, with a greater reduction in survival seen in patients diagnosed with depression and/or treated with anti-depressants. Multi-variable analysis adjusting for age, sex, NYHA class, cardiac risk factors and laboratory parameters found that treatment with anxiolytics remained a predictor of mortality even when adjusting for depression. Depression combined with anxiolytic treatment was predictive of increased mortality, and treatment with anxiolytics alone, depression alone and anxiolytic treatment together with depression were each associated with an increased hazard ratio for a composite outcome of death and hospitalization. Conclusions: In this real-world study of HF patients, both treatment with anxiolytics and depression were associated with increased mortality, and anxiolytic therapy remained a predictor of mortality when adjusting for depression. Treatment of anxiety together with depression was associated with the highest risk of mortality. Safe and effective treatment for anxiety and depression is warranted to alleviate the detrimental impact of these disorders on quality and of life and adverse events.

## 1. Background

Psychosocial factors including lack of social support and depression are associated with poor outcomes in heart failure (HF) patients. Anxiety disorders are the most prevalent of the mental health disorders, with a lifetime risk of 33.7% [1]. In individuals over 65 years of age, the 12-month prevalence of anxiety disorders is 7.6% [1] and anxiety is associated with increased health care utilization [2]. HF patients have rates of anxiety disorders above those found in the general population, with a recent meta-analysis involving 26,366 HF patients reporting that 13.1% of patients surveyed fulfilled diagnostic criteria for an anxiety disorder, and 29% reported clinically significant levels of anxiety [3]. Patients suffering from anxiety most commonly turn to primary care providers for treatment [4], but anxiety is frequently underdiagnosed and under-reported in the primary care setting [5], with similar findings reported in Israel [6]. A review of over 8600 Medicare patients prescribed psychotropic medications between 2005–2015 found that more than 60% were not given a psychiatric diagnosis [7], in particular when primary care providers prescribed the medication. In this study, 51.3% of anxiolytics and 56.5% of anti-depressant prescriptions were given without a psychiatric diagnosis. Anxiety may be under-reported and underdiagnosed in HF patients as well. The signs and symptoms of anxiety such as shortness of breath may overlap with those of HF, further confounding this issue.

Few studies have explored the effect of anxiety and the treatment of anxiety on outcomes in HF patients [8,9,10,11,12], and under-reporting of anxiety further complicates the assessment. This study identified HF patients from a large health maintenance organization database who were prescribed psychotropic medications related to anxiety and/or depression or diagnosed with depression and followed them for HF outcomes. The purpose of the present study was to examine the impact of treatment of anxiety with and without depression on clinical outcomes in a real-world cohort of patients with HF. 

## 2. Methods

Clalit Health Services is the largest health maintenance organization (HMO) in Israel. It has a central computerized database of all members which includes demographics, comprehensive clinical data, diagnoses, and all laboratory data from the single centralized laboratory of the HMO. We identified and retrieved electronically from the computerized database all members in the Jerusalem region with a diagnosis of HF as coded by the database using the International Classification of Diseases, Ninth Revision (ICD-9) code 428. Data was retrieved from January 2017. There were 7106 patients identified with a diagnosis of HF. Patients who were prescribed benzodiazepines other than brotizolam between January 2017 and December 2017 were identified as patients with treated with anxiolytics. Anxiety and insomnia are the primary reasons for the prescription of benzodiazepines, despite the fact that guidelines suggest the use of selective serotonin reuptake inhibitors (SSRIs) or cognitive-behavioral therapy as the first line of therapy [13,14].

Patients who were diagnosed with depression and/or prescribed selective serotonin reuptake inhibitors (SSRIs), serotonin–norepinephrine reuptake inhibitors, serotonin modulator and stimulators, serotonin antagonist and reuptake inhibitors, norepinephrine reuptake inhibitors, norepinephrine–dopamine reuptake inhibitors, tricyclic antidepressants, tetracyclic antidepressants, or monoamine oxidase inhibitors between January 2017 and December 2017 were identified as suffering from depression. Reliable diagnoses of anxiety were not available in this database. Patients that did not survive until January first 2018 (*n* = 813) were excluded from the analysis (landmark analysis.) Patients were then followed for clinical events including cardiovascular hospitalizations and death from January 2018 until January 2019. 

We also performed an additional analysis using anxiolytic therapy and depression/anti-depressant therapy as time-dependent variables. In this analysis we included the entire cohort (n = 7106) and the anxiolytic or anti-depressant therapy variables included any therapy during the entire follow-up of 2 years (2017–2018). These variables were included as time-dependent variables in the Cox regression models.

The type of HF (HF with reduced ejection fraction (HFrEF) and HF with preserved ejection fraction (HFpEF) was based on a documented specific diagnosis and was provided by the treating physician in 67% of the patients. The diagnosis of the remaining patients was ‘Heart failure, unspecified’. All hospitalizations in cardiac and internal medicine departments were retrieved and analyzed. Data on mortality was retrieved from the National Census Bureau. The Institutional Committee for Human Studies of Clalit Health Services, approved the study protocol.

Biochemical analyses were performed at the HMO single centralized core laboratory with routine standardized methodologies on fresh samples of blood or urine obtained after an overnight fast. The laboratory is authorized to perform tests according to the international quality standard ISO-9001.

SPSS version 17.0 for Windows (SPSS Inc., Chicago, IL, USA) and R Statistical Software version 3.0.1 for Windows (R Development Core Team, Vienna, Austria) were used for the analyses. Comparison of the clinical characteristics was performed using the Mann-Whitney U test for continuous variables and the Chi-Square Test for categorical variables. Clinical predictors were transformed where appropriate. Log (10) was used for logarithmic transformations with the exception of estimated glomerular filtration rate (eGFR)for which a square root transformation was used. Follow-up time was calculated using Kaplan-Meier estimate of potential follow-up. Kaplan-Meier curves, with the log-rank test, were used to compare survival according to treatment for anxiety, depression or combined treatment for anxiety and depression. Multivariate Cox proportional hazards regression analysis was used to evaluate independent variables that determined survival. Parameters included in the multivariate Cox regression analysis incorporated age, gender and other clinically significant parameters as well as clinical or laboratory parameters that were significant on univariable analysis with the addition of significant drug therapy in separate models. Proportionality assumptions of the Cox regression models were evaluated by log–log survival curves and with the use of Schoenfeld residuals. An evaluation of the existence of confounding or interactive effects was made between variables and their possible collinearity. A *p* value of < 0.05 was considered statistically significant.

## 3. Results

### 3.1. Clinical Parameters

The study cohort included 6293 HF patients. The characteristics of the patients are presented in Table 1, including demographics, co-morbidities and laboratory values.

Of these patients, 33.9% received pharmacologic treatment for anxiety and/or depression: 15.1% of patients had been prescribed anxiolytics (benzodiazepines,) 26.8% had been prescribed anti-depressant medication, and combined treatment was seen in 8.1% of patients. Patients treated with benzodiazepines and depressed patients were more likely to be older, living on their own without a spouse, and more likely to suffer from comorbidities such as hypertension, atrial fibrillation, prior cerebrovascular event and peripheral vascular disease. No significant differences were seen between patients treated or not treated with benzodiazepines or depressed vs non-depressed patients on anthropomorphic parameters such as BMI, heart rate or blood pressure. The prevalence of treatment for both anxiety and depression increased as the severity of HF increased. (Figure 1), and more women were treated for both anxiety and depression. Of the patients treated with psychotropic medications, 43% of patients prescribed anxiolytics were also prescribed anti-depressants, whereas only 10% of patients prescribed anti-depressants were also prescribed anxiolytic medications.

### 3.2. Treatment for Anxiety, Depression and Clinical Outcomes

Over the one year of follow up, pharmacologic treatment with benzodiazepines was associated with reduced survival (Figure 2A) and reduced event-free survival (Figure 2B) when compared to patients who were not similarly treated. Similar findings with an even more prominent reduction in survival were seen in patients with depression and/or treated with anti-depressant medications compared to patients who were not treated (Figure 2C,D) The curves diverged almost immediately and continued to diverge over the duration of follow-up. No significant difference was seen in survival between patients with depression without medication treatment, with medication treatment, or those who received anti-depressants without a reported diagnosis (Figure 3).

Multivariable analysis (Table 2 and Table 3) adjusting for age, NYHA class, depression and other predictors of mortality on Cox regression analysis showed that treatment for anxiety with benzodiazepines remained predictive of increased mortality, with hazard ratio (HR) of 1.23 (Confidence intervals (CI) 1.0–1.48, *p* = 0.03). Multivariable analysis of depression and/or treatment for depression was also predictive of increased mortality, with a higher HR of 1.49 (CI 1.27–1.75, *p* < 0.001.).

Parameters that were included in the main multivariable analysis model were age, gender, NYHA functional class, diabetes, hypertension, ischemic heart disease, atrial fibrillation, log-transformed body mass index, log-transformed serum urea levels, square root-transformed estimated glomerular filtration rate, hemoglobin, serum sodium, treatment for anxiety and depression status.

Parameters that were included in the multivariable and drugs analysis included the above parameters and the drug treatment with angiotensin-converting enzyme inhibitor/angiotensin receptor blocker/sacubitril-valsartan, beta blocker, furosemide, spironolactone and aspirin.

Parameters included in the additional multivariable analyses were the above parameters with the substitution of treatment for anxiety and depression with a parameter that included either or both.

### 3.3. Comorbid Suspected Anxiety and Depression and Clinical Outcomes

Demographic characteristics of patients with treatment for anxiety alone, depression alone and co-morbid anxiety treatment and depression are presented in Appendix A. Comorbid anxiolytic treatment and depression or either group alone was associated with reduced survival as well as reduced event free survival from death and cardiovascular hospitalizations (Figure 4A,B). HF patients with anxiolytic treatment and depression had reduced survival compared to those with benzodiazepine treatment alone or depression alone, after adjustment for significant parameters (Table 3, additional analysis). Cox regression analysis demonstrated that co-morbid depression combined with treatment with anxiolytics was associated with the highest risk of mortality (HR 1.81, CI 1.44–2.28, *p* < 0.001) (Table 3, additional analysis). The combination was also associated with increased composite outcome of mortality and hospitalizations.

### 3.4. Analysis of Anxiolytics and Depression/Antidepressant Therapy as Time-Dependent Variables

We performed an additional analysis using the parameters anxiolytics and depression/antidepressant therapy as time-dependent variables. This analysis demonstrated that anxiolytics as well as depression/antidepressant therapy were significant independent predictors of increased mortality as well as increased combined end-point of death and cardiovascular hospitalizations (Appendix A). HF patients treated with anxiolytic therapy combined with depression/antidepressant therapy had the highest risk of mortality (HR 2.05 CI 1.72–2.44, *p* < 0.001;) compared to those with anxiolytic therapy alone or depression/antidepressant therapy alone, after adjustment for significant parameters (Appendix A, additional analysis).

## 4. Discussion

In this cohort of real-world HF patients, the prevalence of anxiolytic treatment and depression increased with worsening HF, and 33.9% of patients were diagnosed with depression or received pharmacologic treatment for anxiety and/or depression. Pharmacologic treatment with benzodiazepines was associated with decreased one-year survival compared to untreated individuals, with a greater reduction in survival seen in patients with depression and/or patients receiving treatment for depression; on multivariable analysis, treatment with anxiolytics remained a predictor of mortality, even when adjusting for depression. Patients with depression who were also treated with benzodiazepines demonstrated the largest reduction in survival. Multi-variable analysis adjusting for age, sex, NYHA class, cardiac risk factors and laboratory parameters found that depression together with treatment with benzodiazepines was most predictive of increased mortality. These findings are parallel to those seen in other medical conditions, as anxiety has been associated with increased mortality in cancer patients [15] and in patients with COPD [16], and population studies have found that there is an average of 7.5 excess years of life lost in men with anxiety and 6.3 excess years of life lost in women with anxiety [17]. Anxiety and depression frequently co-exist [18]; patients with major depression report high rates of co-occurring anxiety disorders, and anxiety disorders are more likely to precede the onset of depression rather than the reverse. The combination is associated with increased disability [19]. The co-morbidity may result from overlapping common causes, and the effects of one disorder on the other [20].

Anxiety disorders have been generally accepted as a psychosocial risk factor for coronary heart disease [21], although the role of anxiety independent of depression remains controversial [22]. Patients diagnosed with cardiovascular disease have been found to have increased prevalence of anxiety disorders compared to the general population, and an increased lifetime rates of anxiety disorders [23,24]. Generalized anxiety disorder in outpatients has been associated with increased risk of major adverse cardiac events [25], and panic disorder has been associated with a four-fold risk of coronary heart disease in post-menopausal women [26]. Myocardial infarction patients with co-morbid anxiety have been reported to have a 36% increase in adverse cardiac events and a 47% increase in all-cause mortality [27], but this finding may be confounded by the high incidence of co-morbid depression [28]. The 2016 European Guidelines on cardiovascular disease prevention in clinical practice recommends screening and treatment of both anxiety and depression [21] so as to improve quality of life, although evidence is lacking that treatment of depression and/or anxiety will reduce mortality.

Limited outcomes data are available in HF patients with anxiety. A systematic/review meta-analysis in 2016 found 2 studies [8,29] involving 402 patients that looked at mortality, both of which reported non-significant findings. In that study, the authors were unable to draw conclusions about the association between anxiety and adverse events in HF due to a lack of studies of sufficient quality [12]. Suzuki et al. reported on 221 patients with HF and found an increased risk of death or hospitalization in patients with co-morbid depression and anxiety, but not in either parameter alone [30]. In the Opera-HF study, 779 patients who were admitted for HF were assessed for depression and anxiety, and moderate-severe levels of both parameters were associated with an increased risk of hospitalization and recurrent events. This real-world study with a large HMO cohort suggests that treatment with anxiolytics even in the absence of depression conveys an increased risk of mortality. Depression combined with treatment for anxiety conveys a significant risk for increased mortality in HF patients that exceeds the risk of treatment of depression alone or treatment of anxiety alone.

Increased risk to HF patients in the setting of the anxiety is likely to be multi-factorial. Long term increases in the perception of threat associated with HF in anxious individuals may lead to prolonged activation of the biological stress response and associated sympathetic activation. Anxiety is associated with reduced vagal tone and elevated sympathetic tone in heart rate variability studies [31]. Patients with post-traumatic stress disorder [32,33], panic disorder, and generalized anxiety disorder have higher circulating levels of catecholamines [33]. Similarly, increases in relative muscle sympathetic nerve activity burst amplitude are augmented during acute mental and physiological stress and in anticipation of stress in patients with chronic anxiety [34]. Clinically anxious individuals have also been found to have higher levels of IL-6 [35] and lower levels of morning cortisol, even in the absence of depression [35]. Anxiety is associated with lesser performance of physical activity and lower educational levels, but unlike depression, has not been associated with decreased adherence to pharmacologic treatment [36], although anxious patients are less likely to be adherent to cardiac rehabilitation programs [37]. Anxiety is also associated with sleep disturbance, which may itself present a risk factor for poor outcomes in HF.

The findings of this study may possibly be mediated by adverse effects of anxiolytic medications, rather than the underlying disorders. Large epidemiologic studies have found an association between the use of benzodiazepines and increased mortality. Anxiolytics have been associated with increased risk of falls [38], traffic accidents [39], pneumonia [40], depression, and possibly certain cancers [41]. Wu et al. followed 7419 post MI patients from a Taiwanese registry, and found a j-shaped distribution of outcomes: patients using low doses of benzodiazepines had lower rates of sudden death, cardiovascular mortality and HF hospitalizations, but at higher doses the benzodiazepines were no longer protective and were associated with an increased rate of sudden death [42]. A study of HF patients with insomnia found an increased risk of rehospitalization and a trend toward increased mortality in patients treated with benzodiazepines rather than non-benzodiazepine Z-drugs [43]. Further studies are necessary to clarify the effects of benzodiazepines in the HF population, and to elucidate the impact of anxiety on HF patients.

Our data on depression are similar to the many published studies that find increased mortality in patients with HF and depression. Controversy exists as to the efficacy and safety of anti-depressant therapy in HF patients [44]. The three randomized trials of SSRIs failed to show efficacy in HF patients; no differences in death or hospitalization were seen in these studies [45,46,47]. A network meta-analysis including 2 of these randomized studies and an additional 6 observational studies found an increased risk of all-cause mortality in HF patients taking anti-depressants, but there was no specific medication class that was associated with increased mortality [48]. and the predominance of observational studies complicates the assessment of the impact of the medication or the underlying depression. Our data are similar to those of Brouwers et al., who found that patients with depression, and patients prescribed antidepressant medications with and without depression all had significantly increased mortality compared to non-depressed, non-treated individuals, but no statistically significant difference was found between the depressed group and the groups treated with anti-depressants, whether or not they had been given a diagnosis of depression [49]. Many of the SSRIs have been associated with QT prolongation, and reports from post-marketing surveillance suggests that higher doses of specific serotonin reuptake inhibitors may be associated with torsades de pointes [50]. A small systematic review of SSRI use looking at 1148 depressed patients after myocardial infarction did not find evidence of increased mortality in patients who received SSRIs [51]. A systematic review of non-cardiac patients found observational low-quality studies reporting evidence of increased sudden death associated with higher doses of tricyclic anti-depressants and specific selective serotonin and norepinephrine reuptake inhibitors such as venlaxafine. In addition, they reported an increase in the risk of all-cause mortality in patients taking anti-depressants that prolong the QT interval [52]. HF patients are frequently prescribed anti-arrhythmics and are at risk of polypharmacy and of electrolyte disturbances, which may further increase the risk of potentially lethal arrhythmias in the setting of anti-depressant therapy. Of note, the frequency of filling prescriptions for anti-depressants and benzodiazepines increases as the patient approaches death and may reflect increased discomfort as death approaches rather than causation [53]. This occurs despite studies that suggest that therapy with SSRIs is not effective in the treatment of depression in HF patients.

### Limitations

Several potential limitations of this study merit consideration. The present study was an observational study. Data regarding clinical parameters and drug therapy was based on a digitized database. Although this database was validated and found to be highly accurate, not all data could be verified. This study evaluated treatment with benzodiazepines and not anxiety. The diagnosis of depression was based on a clinical impression or medication treatment, and not based on a formal psychological evaluation or psychological testing. This was done due to database limitations but may have led to inclusion of inappropriate patients including those suffering from insomnia or chronic pain. As more than 40% of patients may not receive pharmacologic treatment for significant anxiety or depression [54], this may have led to under-assessment of the burden of anxiety and depression in this population. This study did not evaluate other evidence-based therapies for anxiety and depression, including cognitive-behavioral therapy and exercise therapies [55]. While we tried to adjust for clinically relevant parameters, not all clinical parameters were available, and it is impossible to adjust for all variables that may affect outcome. In addition, the cohort was a community-based cohort, and the findings may not be applicable in more advanced or hospital-based HF cohorts.

In conclusion, in this real-world study of HF patients, pharmacologic treatment with anxiolytics was associated with increased mortality, even when adjusting for the diagnosis or treatment of depression. The combination of treatment for anxiety and depression/treatment for depression was associated with the steepest increase in mortality. Given the high rates and adverse outcomes of anxiety, depression and the combination in HF patients, effective screening and further elucidation of safe and effective treatment for anxiety and depression is warranted to alleviate the detrimental impact of these disorders on quality and of life and adverse events.

## Figures and Tables

**Figure 1 jcm-09-03967-f001:**
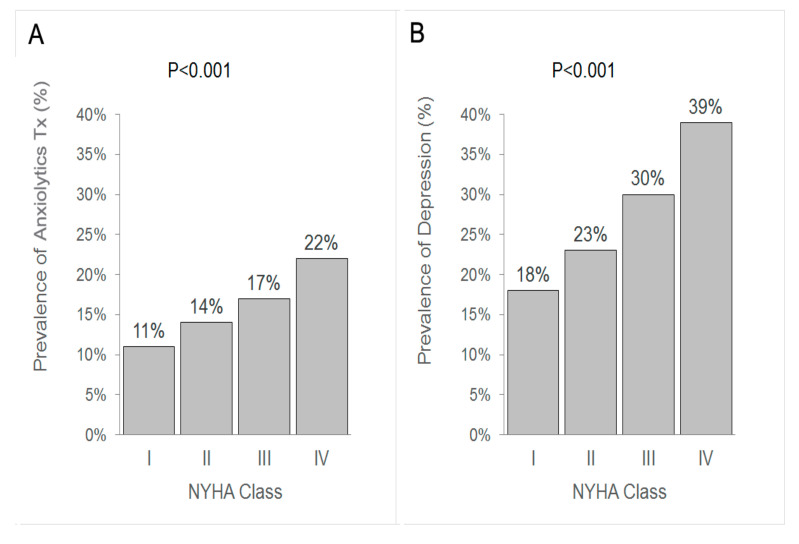
The relationship between the New York Heart Association Class and the prevalence of anxiolytic therapy (**A**,**B**) depression.

**Figure 2 jcm-09-03967-f002:**
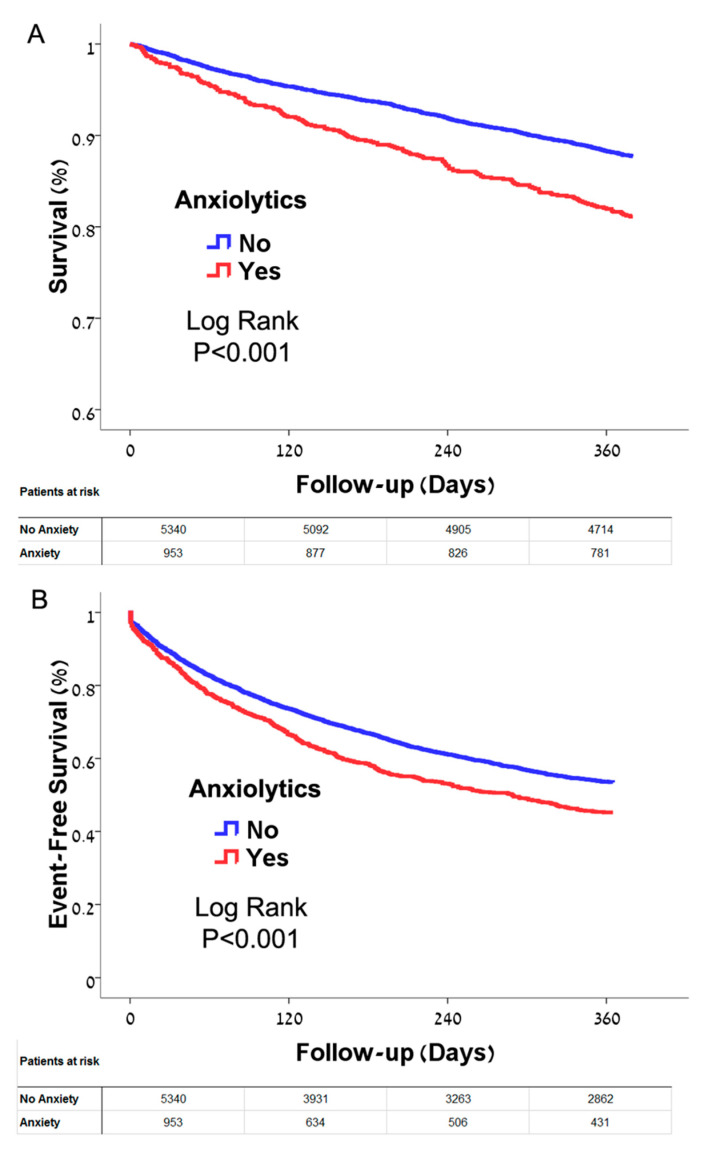
(**A**) Kaplan-Meier estimate of survival in heart failure patients with and without treatment with anxiolytics. Treatment of anxiety with benzodiazepines was associated with increased mortality (*p* < 0.001). Log rank test was used to test significance. (**B**) Kaplan-Meier estimate of event-free survival (mortality and cardiovascular hospitalizations) in heart failure patients with and without anxiolytics. Treatment of anxiety with benzodiazepines was associated with reduced event free survival (*p* < 0.001). Log rank test was used to test significance. (**C**) Kaplan-Meier estimate of survival in heart failure patients with and without depression. Depression and/or treatment with anti-depressants was associated with increased mortality (*p* < 0.001). Log rank test was used to test significance. (**D**) Kaplan-Meier estimate of event-free survival (mortality and cardiovascular hospitalizations) in heart failure patients with and without depression. Treatment with anti-depressants was associated with reduced event-free survival (*p* < 0.001). Log rank test was used to test significance.

**Figure 3 jcm-09-03967-f003:**
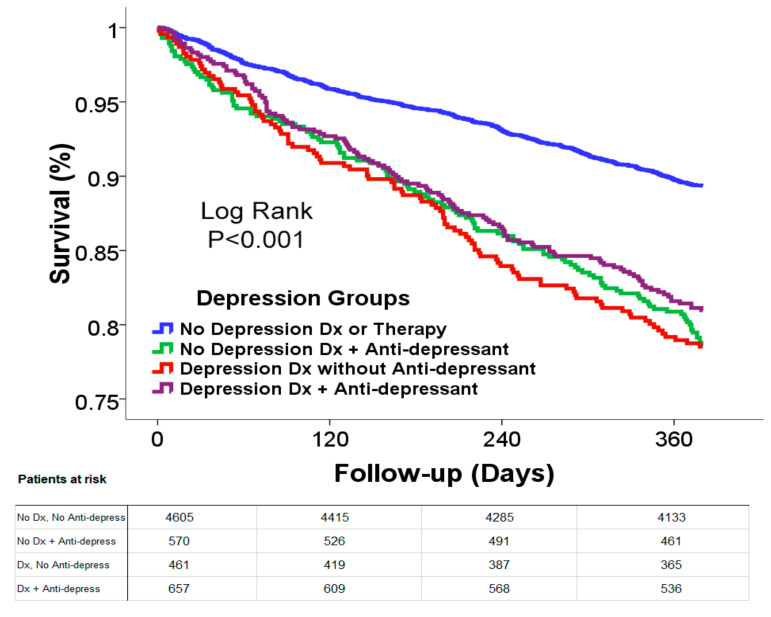
Kaplan-Meier estimate of survival in heart failure patients stratified according to diagnosis of depression and anti-depressant therapy. The groups that were with a diagnosis and/or anti-depressant therapy had reduced survival compared to patients without a diagnosis or therapy for depression (Log rank *p* < 0.001).

**Figure 4 jcm-09-03967-f004:**
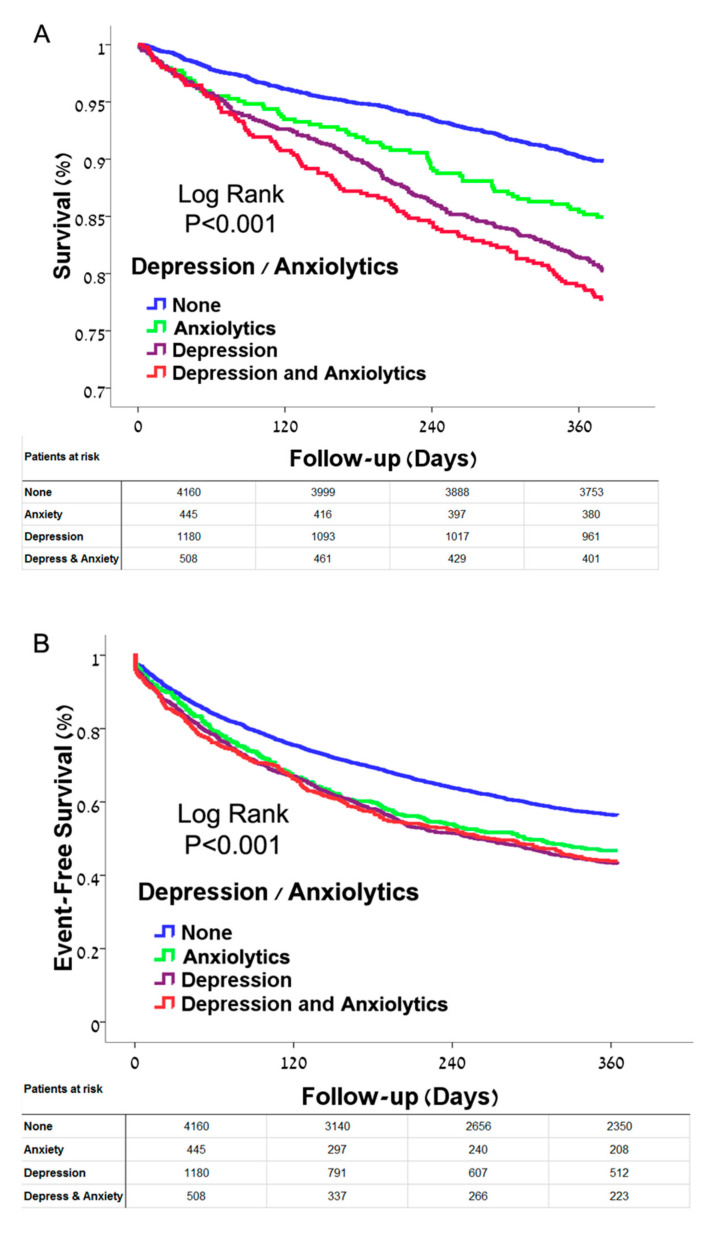
(**A**) Kaplan-Meier estimate of survival in heart failure patients with treatment for anxiety only, depression only, treatment for anxiety together with depression and not-treated. Reduced survival was seen in patients treated with anxiolytics only compared to non-treated patients, with a further reduction in survival seen in patients treated for depression only. The greatest reduction in survival was seen in patients with depression who were also treated with anxiolytics (*p* < 0.001). Log rank test was used to test significance. (**B**) Kaplan-Meier estimate of event-free survival (mortality and hospitalizations) in heart failure patients with treatment for anxiety only, depression only, treated for anxiety together with depression and non-depressed/non-treated. Reduced survival was seen in each group of treated patients compared to non-depressed, non-treated patients (*p* < 0.001); no significant differences in event free survival were seen between depressed group and the anxiolytic groups or the co-morbid group. Log rank test was used to test significance.

**Table 1 jcm-09-03967-t001:** Demographics and clinical characteristics of patients with heart failure according to depression or anxiety.

Variable	No. Anxiolytic (*N* = 5340)	Anxiolytics (*N* = 953)	*p* Value	No. Depression (*N* = 4605)	Depression (*N* = 1688)	*p* Value	Total (*N* = 6293)
Age (Years)	75 (65–84)	80 (69–87)	<0.001	74 (64–83)	81 (72–88)	<0.001	76 (66–84)
Men	2925 (55)	416 (44)	<0.001)	2664 (58)	677 (40)	<0.001	3341 (53)
Social Status (living with spouse)	2576 (49)	380 (40)	<0.001	2371 (53)	585 (35)	<0.001	2956 (48)
NYHA Class III/IV	1465 (36)	302 (44)	<0.001	1224 (34)	543 (45)	<0.001	1767 (37)
HF Type							
Reduced ejection fraction	1493 (28%)	250 (26%)	<0.001	1320 (29%)	423 (25%)	<0.001	1743 (28%)
Preserved ejection fraction	1992 (37%)	401 (42%)		1654 (36%)	739 (44%)		2393 (38%)
Not-specified	1855 (35%)	302 (32%)		1631 (35%)	526 (31%)		2157 (34%)
Diabetes mellitus	2852 (53)	480 (50)	0.08	2443 (53)	889 (53)	0.79	3332 (53)
Hypertension	4326 (81)	837 (88)	<0.001	3676 (80)	1487 (88)	<0.001	5163 (82)
Hyperlipidemia	4749 (89)	866 (91)	0.08	4075 (88)	1540 (91)	0.002	5615 (89)
Ischemic Heart Disease	3495 (65)	626 (66)	0.89	3046 (66)	1075 (64)	0.07	4121 (65)
Prior Myocardial Infarction	2256 (42)	391 (41)	0.48	1989 (43)	658 (39)	0.003	2647 (42)
Prior coronary bypass surgery	101 (2)	11 (1)	0.11	90 (2)	22 (1)	0.08	112 (2)
Atrial fibrillation	1935 (36)	397 (42)	0.001	1640 (36)	692 (41)	<0.001	2332 (37)
Prior Stroke/transient ischemic attack	1168 (22)	247 (26)	0.006	864 (19)	551 (33)	<0.001	1415 (22)
Peripheral vascular disease	716 (13)	159 (17)	0.007	570 (12)	305 (18)	<0.001	875 (14)
Chronic obstructive lung disease	1104 (21)	229 (24)	0.02	941 (20)	392 (23)	0.02	1333 (21)
Charlson Comorbidity Score	6.0 (5.0–7.0)	7.0 (5.0–8.0)	<0.001	6.0 (4.0–7.0)	7.0 (5.0–8.0)	<0.001	6.0 (5.0–8.0)
Malignancy	1060 (20)	252 (26)	<0.001	900 (20)	412 (24)	<0.001	1312 (21)
Dementia	596 (11)	199 (21)	<0.001	329 (7)	466 (28)	<0.001	795 (13)
Dialysis	256 (5)	44 (5)	0.81	231 (5)	69 (4)	0.13	300 (5)
Body mass index (kg/m^2^)	29 (26–33)	29 (25–33)	0.06	29 (26–33)	29 (25–33)	0.07	29 (25–33)
Pulse (beats per minute)	72 (65–80)	72 (64–80)	0.03	72 (64–80)	72 (64–80)	0.25	72 (64–80)
Systolic blood pressure (mmHg)	128 (118–138)	128 (117–139)	0.65	128 (118–138)	128 (116–139)	0.72	128 (118–139)
Diastolic blood pressure (mmHg)	72 (65–80)	70 (63–79)	0.21	72 (65–80)	70 (63–80)	0.14	72 (65–80)
**Selected Laboratory Data**	**No. Anxiolytic (*N* = 5340)**	**Anxiolytics (*N* = 953)**	***p* Value**	**No. Depression (*N* = 4605)**	**Depression (*N* = 1688)**	***p* Value**	**Total (*N* = 6293)**
Estimated glomerular filtration rate (mL/min per 1.73 m^2^) *	71 (51–94)	68 (48–88)	0.001	73 (52–95)	66 (48–88)	<0.001	71 (51–93)
Urea (mg/dL)	44 (34–62)	47 (34–64)	0.05	44 (33–61)	48 (35–66)	<0.001	45 (34–62)
Sodium (mEq/L)	140 (138–142)	140 (138–142)	0.52	140 (138–142)	140 (138–142)	0.30	140 (138–142)
Potassium (mEql/L)	13 (12–14)	13 (11–14)	<0.001	13 (12–14)	12 (11–14)	<0.001	13 (12–14)
Hemoglobin (g/dL)	4.6 (4.3–4.9)	4.6 (4.3–4.9)	0.17	4.6 (4.3–5.0)	4.6 (4.3–4.9)	0.08	4.6 (4.3–4.9)
Red Cell Distribution Width (%)	15 (14–16)	15 (14–16)	<0.001	15 (14–16)	15 (14–16)	<0.001	15 (14–16)
Glucose (mg/dL)	107 (94–134)	104 (93–126)	<0.001	108 (95-135)	103 (92-129)	<0.001	106 (94-133)
Hemoglobin A1c (%)	6.1 (5.6–7.2)	6.0 (5.5–6.8)	<0.001	6.1 (5.6–7.2)	6.0 (5.6–6.9)	<0.001	6.1 (5.6–7.1)
Uric Acid (mg/dL)	6.2 (5.1–7.5)	6.2 (5.0–7.5)	0.33	6.2 (5.1–7.5)	6.2 (5.1–7.5)	0.61	6.2 (5.1–7.5)
TSH (mIU/L)	2.2 (1.5–3.3)	2.4 (1.5–3.6)	0.02	2.2 (1.5–3.3)	2.3 (1.5–3.4)	0.52	2.2 (1.5–3.3)
Transferrin Saturation (%)	17 (12–23)	16 (12–22)	0.17	17 (12–23)	17 (12–23)	0.44	17 (12–23)
Ferritin (ng/mL)	81 (38–166)	76 (36–163)	0.53	81 (37–165)	78 (39–169)	0.68	80 (38–166)
Triglycerides (mg/dL)	120 (89–170)	122 (91–169)	0.70	120 (88–170)	122 (91–168)	0.60	121 (89–169)
Low-density lipoprotein (mg/dL)	82 (65–105)	86 (65–108)	0.13	82 (65–105)	83 (66–108)	0.75	83 (65–106)
Albumin (g/dL)	4.0 (3.7–4.2)	3.9 (3.6–4.1)	<0.001	4.0 (3.7–4.2)	3.8 (3.5–4.1)	<0.001	3.9 (3.7–4.2)
C-Reactive Protein (mg/dL)	0.6 (0.2–1.4)	0.6 (0.2–1.8)	0.15	0.5 (0.2–1.4)	0.6 (0.2–1.6)	0.20	0.6 (0.2–1.5)
**Medication**	**No. Anxiolytic (*N* = 5340)**	**Anxiolytics (*N* = 953)**	***p* Value**	**No. Depression (*N* = 4605)**	**Depression (*N* = 1688)**	***p* Value**	**Total (*N* = 6293)**
ACE-I/ARB/ARNI	4156 (78)	747 (78)	0.7	3651 (79)	1252 (74)	<0.001	4903 (78)
Beta blockers	3967 (74)	730 (77)	0.13	3462 (75)	1235 (73)	0.1	4697 (75)
Spironolactone	3421 (64)	703 (74)	<0.001	2889 (63)	1235 (73)	<0.001	4124 (66)
Furosemide	1866 (35)	370 (39)	0.02	1614 (35)	622 (37)	0.19	2236 (36)
Thiazide	752 (14)	115 (12)	0.1	641 (14)	226 (13)	0.59	867 (14)
Digoxin	318 (6)	65 (7)	0.3	263 (6)	120 (7)	0.04	383 (6)
Amiodarone	817 (15)	169 (18)	0.06	758 (16)	228 (14)	0.004	986 (16)
Aspirin	153 (3)	32 (3)	0.41	129 (3)	56 (3)	0.28	185 (3)
New oral anticoagulants **	2991 (56)	526 (55)	0.64	2640 (57)	877 (52)	<0.001	3517 (56)
Vitamin K antagonists	223 (4)	28 (3)	0.07	199 (4)	52 (3)	0.03	251 (4)
Anti-Depressants	816 (15)	411 (43)	<0.001	0	1227 (73)		1227 (19)
Anxiolytics	0	953 (100)		445 (10)	508 (30)	<0.001	953 (15)

Data is presented as median (inter-quartile range) for continuous variables and counts (percentages) for categorical variables. *p* value by the Kruskal Wallis Test for continuous variables and the Chi-Square Test for categorical variables. Diabetes mellitus defined as fasting plasma glucose ≥126 mg/dL or glucose lowering treatment, hypertension as blood pressure >140/90 mmHg measured on several occasions or anti-hypertensive treatment and hyperlipidemia as low density lipoprotein >130 mg/dL, fasting serum triglycerides >200 mg/dL or lipid lowering treatment. * Estimated Glomerular Filtration Rate was calculated using the modified Modification of Diet in Renal Disease (MDRD) equation (175 * serum creatinine^–1.154^ * age^–0.203^. For females a correction factor is used multiplying by 0.742). ** Dabigatran, Rivaroxaban or Apixaban.

**Table 2 jcm-09-03967-t002:** Predictors of mortality by Cox regression analysis.

	Univariable		Multivariable	
	Hazard Ratio (95% CI)	*p* Value	Hazard Ratio (95% CI)	*p* Value
Age (years)	1.06 (1.05–1.07)	<0.001	1.04 (1.03–1.05)	<0.001
Gender (Male)	0.80 (0.70–0.92)	<0.001	1.26 (1.06–1.48)	0.007
NYHA III/IV	1.99 (1.68–2.36)	<0.001	1.33 (1.11–1.60)	0.002
Diabetes Mellitus	1.16 (1.01–1.33)	0.003	1.19 (1.01–1.40)	0.03
Hypertension	2.55 (1.99–3.26)	<0.001	1.24 (0.94–1.64)	0.13
Ischemic Heart Disease	1.04 (0.90–1.20)	0.33	0.87 (0.74–1.03)	0.11
Atrial Fibrillation	1.80 (1.57–2.06)	<0.001	1.44 (1.24–1.68)	<0.001
Body mass index * (kg/m^2^)	0.13 (0.06–0.29)	<0.001	0.21 (0.08–0.51)	<0.001
Urea (mg/dL) *	9.18 (6.80–12.39)	<0.001	3.85 (2.17–6.85)	<0.001
eGFR **(mL/min per 1.73 m^2^)	0.84 (0.81–0.87)	<0.001	1.00 (0.94–1.07)	0.95
Sodium (mEq/L)	0.95 (0.93–0.97)	<0.001	0.97 (0.95–0.99)	0.007
Hemoglobin (g/dL)	0.79 (0.76–0.82)	<0.001	0.89 (0.85–0.93)	<0.001
Treatment for Anxiety	1.60 (1.36–1.89)	<0.001	1.23 (1.02–1.48)	0.03
Depression	2.04 (1.78–2.34)	<0.001	1.49 (1.27–1.75)	<0.001

Data is presented as hazard ratio (95% confidence interval), *p* value. * Log-transformed ** Square root-transformed.

**Table 3 jcm-09-03967-t003:** Hazard ratio for clinical outcome according to anxiety and depression status by Cox regression analysis.

	Univariable		Multivariable		Multivariable and Drugs	
	Hazard Ratio (95% CI)	*p*-Value	Hazard Ratio (95% CI)	*p*-Value	Hazard Ratio (95% CI)	*p*-Value
Main Analysis						
Death						
Anxiolytics	1.60 (1.36–1.89)	<0.001	1.23 (1.02–1.48)	0.03	1.24 (1.03–1.50)	0.02
Depression	2.04 (1.78–2.34)	<0.001	1.49 (1.27–1.75)	<0.001	1.46 (1.25–1.72)	<0.001
Death and cardiovascular hospitalization						
Anxiolytics	1.28 (1.16–1.41)	<0.001	1.08 (0.97–1.20)	0.17	1.05 (0.95–1.17)	0.35
Depression	1.42 (1.32–1.54)	<0.001	1.20 (1.10–1.31)	<0.001	1.19 (1.09–1.30)	<0.001
Additional Analysis						
Death						
None	1.0 (Reference)	<0.001	1.0 (Reference)	<0.001	1.0 (Reference)	<0.001
Anxiolytics (Alone)	1.53 (1.18–1.98) 0.001		1.26 (0.95–1.67) 0.10		1.28 (0.97–1.70) 0.08	
Depression (Alone)	2.05 (1.75–2.41) <0.001		1.51 (1.26–1.80) <0.001		1.48 (1.24–1.78) <0.001	
Depression and Anxiolytics	2.36 (1.91–2.90) <0.001		1.81 (1.44–2.28) <0.001		1.80 (1.43–2.27) <0.001	
Death and cardiovascular hospitalization						
None	1.0 (Reference)	<0.001	1.0 (Reference)	<0.001	1.0 (Reference)	<0.001
Anxiolytics (Alone)	1.34 (1.17–1.53) <0.001		1.23 (1.06–1.41) 0.005		1.21 (1.04–1.39) 0.01	
Depression (Alone)	1.47 (1.35–1.61) <0.001		1.27 (1.15–1.40) <0.001		1.26 (1.14–1.39) <0.001	
Depression and Anxiolytics	1.46 (1.29–1.65) <0.001		1.20 (1.04–1.37) 0.01		1.15 (1.01–1.32) 0.04	

Data is presented as hazard ratio (95% confidence interval), *p* value.

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
