# Peer review of "Treatment of Heart Failure Patients with Anxiolytics Is Associated with Adverse Outcomes, with and without Depression"

_jcm, 2020, doi:10.3390/jcm9123967_

Round 1

Reviewer 1 Report

In their study, Zwas and colleagues report results that are extremely important for clinicians. They investigated the effect of taking or prescribing benzodiazepines and antidepressants on survival. A real-world cohort of n = 6293 HF patients was investigated, of whom n = 953 (15%) were prescribed benzodiazepines, and n = 1688 antidepressants (27%). The prescription of these drugs was associated with increased mortality even after adjusting for other important risk factors (sex, partnership, and severity of HF). This is an alarming result. The result is particularly concerning because the current evidence shows that in HF patients, antidepressants are ineffective in reducing depression (Angermann et al. 2016, Fraguas et al. 2009, O'Connor et al. 2010) and that antidepressants are very likely to increase mortality in HF patients (He et al. 2020, Biffi et al. 2020). Long-term use of benzodiazepines contradicts all guidelines for the treatment of mental disorders.

Furthermore, the authors revealed that psychotropic drugs were prescribed without an adequate diagnosis. I do not consider all this to be unusual. However, this finding is a scandal that reveals deficits in real-world medical care and systematic stigmatization of emotional disorders. It is, therefore, important that these data will be published.

However, a thorough revision of the manuscript, including the title, is necessary. The title should be changed to something like "Treatment of Heart Failure Patients with Benzodiazepines and Antidepressants is Associated with Increased Mortality". Accordingly, the manuscript should not refer to diagnoses but the prescription or intake of psychiatric drugs. That means that the manuscript has to be revised entirely. A reduction of the length of the manuscript by a more concise presentation should also be aimed for. In the introduction and discussion, the authors should point out firstly, that the current evidence shows that antidepressants do not have demonstrated a favorable effect in HF patients, secondly that antidepressants probably increase mortality of HF patients, thirdly that alternative evidence-based treatment methods exist (Das et al 2019), and fourthly that the lack of diagnostic documentation points to a problematic stigmatization of mentally ill patients and corresponding deficits in medical education.

Minor comment (line 62: diagnosisAnxiety = typo.)

References

Angermann, C.E., Gelbrich, G., Stork, S., Gunold, H., Edelmann, F., Wachter, R., Schunkert, H., Graf, T., Kindermann, I., Haass, M., Blankenberg, S., Pankuweit, S., Prettin, C., Gottwik, M., Bohm, M., Faller, H., Deckert, J., Ertl, G., 2016. Effect of Escitalopram on All-Cause Mortality and Hospitalization in Patients With Heart Failure and Depression: The MOOD-HF Randomized Clinical Trial. Jama 315, 2683-2693.

Biffi, A., Rea, F., Scotti, L., Lucenteforte, E., Vannacci, A., Lombardi, N., Chinellato, A., Onder, G., Vitale, C., Cascini, S., Ingrasciotta, Y., Roberto, G., Mugelli, A., Corrao, G., 2020. Antidepressants and the Risk of Cardiovascular Events in Elderly Affected by Cardiovascular Disease: A Real-Life Investigation From Italy. Journal of clinical psychopharmacology 40, 112-121.

Das, A., Roy, B., Schwarzer, G., Silverman, M.G., Ziegler, O., Bandyopadhyay, D., Philpotts, L.L., Sinha, S., Blumenthal, J.A., Das, S., 2019. Comparison of treatment options for depression in heart failure: A network meta-analysis. J Psychiatr Res 108, 7-23.

Fraguas, R., da Silva Telles, R.M., Alves, T.C., Andrei, A.M., Rays, J., Iosifescu, D.V., Wajngarten, M., 2009. A double-blind, placebo-controlled treatment trial of citalopram for major depressive disorder in older patients with heart failure: the relevance of the placebo effect and psychological symptoms. Contemp Clin Trials 30, 205-211.

He W, Zhou Y, Ma J, Wei B, Fu Y. Effect of antidepressants on death in patients with heart failure: a systematic review and meta-analysis. Heart Fail Rev. 2020 Nov;25(6):919-926.

O'Connor, C.M., Jiang, W., Kuchibhatla, M., Silva, S.G., Cuffe, M.S., Callwood, D.D., Zakhary, B., Stough, W.G., Arias, R.M., Rivelli, S.K., Krishnan, R., 2010. Safety and efficacy of sertraline for depression in patients with heart failure: results of the SADHART-CHF (Sertraline Against Depression and Heart Disease in Chronic Heart Failure) trial. J Am Coll Cardiol 56, 692-699.

Author Response

In the methods section, we did not clearly describe the group identified as patients with depression.  This group consistent of patients for whom there was either a diagnosis of depression or treatment with an anti-depressant. In Table 1 you can see that 73% of the group patients were prescribed an antidepressant, and 23% were diagnosed with depression but not treated. We have revised the methods section to clearly delineate this, and made changes throughout the manuscript to reflect this.

In light of the reviewers’ comments, we performed an additional analysis of our data. Similar to the report s/he cited of Brouwers et al., we found that patients with the diagnosis of depression, patient with depression and anti-depressant treatment, and patients treated with anti-depressants in the absence of a formal diagnosis, all had reduced survival compared with non-depressed, non treated patients, but there were no significant differences between groups. We have included these findings in the paper.

In response to the reviewer’s comments, we have expanded the discussion to include a discussion of treatment efficacy of anti-depressant medications, and further expanded our existing discussion of treatment risks. We have also included a comment as to other evidence-based treatments available in heart failure patients. The paper has been significantly revised.

Reviewer 2 Report

The focus of this manuscript is on the prospective relationship of metal health conditions to medical outcomes in patients with diagnosed heart failure.  While this relationship has previously been reported - especially for depression - the context provided here of a large, nationwide cohort, whereby the electronic health record is leveraged to define cases increase the importance of the manuscript and findings.  The research methodology is straight forward and the analytic plan is largely sound.

At the same time, there are a number of weaknesses in the report.

  1. It is unfortunate that no information regarding mental health conditions - anxiety, depression - is available beyond the prescribing of anxiolytic and/or antidepressant medications.  As noted by the authors, anxiolytics are often prescribed for sleep issues (which are highly prevalent in heart failure), and SSRI medications are often prescribed for anxiety.  Can the authors provide information as to the presence of diagnoses in the health record vs. medication prescription?  Are narrative notes available whereby the authors might randomly select patients with a prescription to ascertain whether there are patient reports of associated symptoms, and the 'hit rate', or the percent of patients prescribed a medication who also have health record evidence of the associated disorder?  Some method for determining the accuracy of their assumption that a prescription for a given medication is evidence of the psychiatric condition is essential.
  2. The description of the research methods can be made more clear with regard to the period of sampling and the period of follow-up.  Given the time frame, it would seem possible to select a given date (e.g., January 2017) and identify all patients who at that time carried a heart failure diagnosis AND had a prescription for the medications of interest, and then to follow those patients for the 2 years of this study.  The authors describe a 1 year period for patients with heart failure who were prescribed the medication(s) in the subsequent year, and the patients were then followed for an additional year, censoring those who died during the year when data were collected.  It is not clear why the authors chose this strategy, since this was essentially a retrospective 'look' at the health record. 
  3. This strategy also creates issues of exposure time - e.g., a person prescribed a target medication in December of 2017 'counts' in the same way that a person prescribed a target med in January of 2017 does, yet, the latter patients has more exposure time with the medication and thus, the mental health condition.
  4. It is furthermore unfortunate, that the authors do not report on patient history of anxiety/depression, which should have been available by looking further back in the health record, even using their method for defining anxiety/depression by medication prescription.  Incorporating data as to history would provide a more complete picture, as heart failure alone can provoke anxiety/depression, and thus the finding here may be more a function of disease severity/experience, than of any underlying mental health condition.
  5. The authors describe - and provide data on - biochemical assays.  When were these assays performed within the 1 year period of data collection?
  6. In the mutivariate analysis, the authors include a range of covariates.  It is unclear why they did not include the Charleson Comorbidity Index in their models.

Author Response

  1. It is unfortunate that no information regarding mental health conditions - anxiety, depression - is available beyond the prescribing of anxiolytic and/or antidepressant medications.  As noted by the authors, anxiolytics are often prescribed for sleep issues (which are highly prevalent in heart failure), and SSRI medications are often prescribed for anxiety.  Can the authors provide information as to the presence of diagnoses in the health record vs. medication prescription?  Are narrative notes available whereby the authors might randomly select patients with a prescription to ascertain whether there are patient reports of associated symptoms, and the 'hit rate', or the percent of patients prescribed a medication who also have health record evidence of the associated disorder?  Some method for determining the accuracy of their assumption that a prescription for a given medication is evidence of the psychiatric condition is essential.

In the methods section, we did not clearly describe the group identified as patients with depression.  This group consistent of patients for whom there was either a diagnosis of depression or treatment with an anti-depressant. In Table 1 you can see that 73% of the group patients were prescribed an antidepressant, and 23% were diagnosed with depression but not treated. We have revised the methods section to clearly delineate this. Unfortunately, we did not have access to diagnoses of anxiety or sleep disorder, or other narrative notes. This is a limitation of the study.

  1. The description of the research methods can be made more clear with regard to the period of sampling and the period of follow-up.  Given the time frame, it would seem possible to select a given date (e.g., January 2017) and identify all patients who at that time carried a heart failure diagnosis AND had a prescription for the medications of interest, and then to follow those patients for the 2 years of this study.  The authors describe a 1 year period for patients with heart failure who were prescribed the medication(s) in the subsequent year, and the patients were then followed for an additional year, censoring those who died during the year when data were collected.  It is not clear why the authors chose this strategy, since this was essentially a retrospective 'look' at the health record. 

We chose this strategy as we wanted to include as many as possible patients that were prescribed psychotropic medications. A non-significant minority of patients were prescribed these medications over the course of the first year (2017) and if we would only include patients that were prescribed these medications at the beginning of 2017, the sample would be far smaller and less representative.  The way we constructed the analysis allowed us to include a large number of patients that were prescribed these medications.

  1. This strategy also creates issues of exposure time - e.g., a person prescribed a target medication in December of 2017 'counts' in the same way that a person prescribed a target med in January of 2017 does, yet, the latter patients has more exposure time with the medication and thus, the mental health condition.

We acknowledge this limitation; however, this issue is also present even if we only include patients prescribed medication at the beginning of 2017.  Our database did not enable us to ascertain the start date of a given medication, which is the only method that would enable us to account for exposure time.

  1. It is furthermore unfortunate, that the authors do not report on patient history of anxiety/depression, which should have been available by looking further back in the health record, even using their method for defining anxiety/depression by medication prescription.  Incorporating data as to history would provide a more complete picture, as heart failure alone can provoke anxiety/depression, and thus the finding here may be more a function of disease severity/experience, than of any underlying mental health condition.

We accept the reviewer’s concerns. In this health system, both anxiety and depression are under-diagnosed, and the diagnosis does not require formal testing or the involvement of a mental health specialist. The diagnosis of depression was available to us, but the diagnosis of anxiety was not. As described in comment 1, we did include those who had a diagnosis of depression with or without medications, and unfortunately this was previously not adequately described in the methods section. We have clarified this issue and updated the methods section regarding the diagnosis of depression.

  1. The authors describe - and provide data on - biochemical assays.  When were these assays performed within the 1 year period of data collection?

The biochemical assays were performed during 2017.

  1. In the mutivariate analysis, the authors include a range of covariates.  It is unclear why they did not include the Charlson Comorbidity Index in their models.

We appreciate this comment. Indeed, the Charlson Index is a predictor of outcome. However, this index includes numerous parameters and including it would necessitate excluding several key parameters that are better represented in the model separately.

Reviewer 3 Report

This is a useful and generally well-presented article. Data analyses are very carefully done. A few issues should be addressed:

1. The authors present numbers of individuals with different cardiovascular medications. Numbers of individuals with particular classes of cardiovascular drugs should be presented also--this is currently missing in the manuscript. Not essential, but of interest also, and possible in these data because of sample size, would be different cardiovascular outcomes based on different medication classes. 

2. The study is based on classifying depression and anxiety based on medications. A fundamental problem with this is that (1) the medications themselves may be having adverse effects or interactions with other medications the patient is taking; and (2) this only captures individuals with depression or anxiety who are medicated and not those not medicated or treated via psychotherapy or other means. This is mentioned in the discussion lines 284-301, but they may want to incorporate some of this caution of interpretation in their labels (see point #3 below) and use of language in the text. 

3. Following from point #2, one way of separating out the effects of medications and/or patients selected to be medicated is to present outcomes for those individuals with diagnoses of depression and/or anxiety who are not medicated. Do the investigators have the ability to determine those individuals with diagnoses of depression and anxiety without taking these medications? If so, a survival curve examining non-medicated depressive and/or anxiety patients would be useful.

4. Based on the issues in point #2, the authors should be careful in using the terms "depression" and "anxiety" since the actual determinants of classification are antidepressant medication and anti-anxiety medication. For this reason, they may want to modify their labels in tables and figures. 

5. Too much data is presented in Table 1 for the reader to absorb and also much of this information is not relevant to the present study. I would recommend deleting most of the laboratory data values.

6. There are conflicting findings in the literature regarding outcomes in patients medicated vs. not medicated for depression, over and above the issue of adverse effects of tricyclic medications. For example, There is conflicting information as to whether taking antidepressants are associated with adverse outcomes in patients with CHD. For example, one analysis from the ENRICHD study of post-MI patients (Taylor et al. Arch Gen Psychiatry) showed better outcomes in those taking SSRI's, whereas the present study and others find adverse effects of antidepressant use in patients with suspected CAD (e,g,  Krantz et al., from the WISE study, Heart; and other studies). The authors do not review the conflicting evidence in the literature based on studies using depression medication as a proxy for depression. In addition, I may have missed it in the lengthy reference list, but the set of SADHEART intitial study as well as the SADHEART-CHF clinical trial of heart failure patients did not show adverse effects outcome in medicated patients. The authors need to consider these conflicting findings in their discussion. 

Author Response

done. A few issues should be addressed:

  1. The authors present numbers of individuals with different cardiovascular medications. Numbers of individuals with particular classes of cardiovascular drugs should be presented also--this is currently missing in the manuscript. Not essential, but of interest also, and possible in these data because of sample size, would be different cardiovascular outcomes based on different medication classes. 

In reponse to the reviewer, an additional supplementary table was added, with the number of individuals with particular classes of cardiovascular drugs.

  1. The study is based on classifying depression and anxiety based on medications. A fundamental problem with this is that (1) the medications themselves may be having adverse effects or interactions with other medications the patient is taking; and (2) this only captures individuals with depression or anxiety who are medicated and not those not medicated or treated via psychotherapy or other means. This is mentioned in the discussion lines 284-301, but they may want to incorporate some of this caution of interpretation in their labels (see point #3 below) and use of language in the text. 

In the methods section, we did not clearly describe the group identified as patients with depression.  This group consistent of patients for whom there was either a diagnosis of depression or treatment with an anti-depressant. In Table 1 you can see that 73% of the group patients were prescribed an antidepressant, and 23% were diagnosed with depression but not treated. We have revised the methods section to clearly delineate this, and made changes throughout the manuscript. Unfortunately, we did not have access to diagnoses of anxiety or sleep disorder, or other narrative notes. This is a limitation of the study

  1. Following from point #2, one way of separating out the effects of medications and/or patients selected to be medicated is to present outcomes for those individuals with diagnoses of depression and/or anxiety who are not medicated. Do the investigators have the ability to determine those individuals with diagnoses of depression and anxiety without taking these medications? If so, a survival curve examining non-medicated depressive and/or anxiety patients would be useful.

We do have access to data regarding the diagnosis of depression, and in light of this reviewer’s and other reviewers’ comments have included a survival curve exploring the outcomes of patients with the diagnosis of depression who were treated and untreated, and those patients treated without a formal diagnosis of depression. We found that patients with the diagnosis of depression, patients with depression and anti-depressant treatment, and patients treated with anti-depressants in the absence of a formal diagnosis, all had reduced survival compared with non-depressed, non-treated patients, but there were no significant differences between groups.

  1. Based on the issues in point #2, the authors should be careful in using the terms "depression" and "anxiety" since the actual determinants of classification are antidepressant medication and anti-anxiety medication. For this reason, they may want to modify their labels in tables and figures. 

As mentioned above, in the methods section, we did not clearly describe the group identified as patients with depression.  This group consistent of patients for whom there was either a diagnosis of depression or treatment with an anti-depressant. We did not have access to the diagnosis of anxiety, and the reviewer’s point is well taken.

  1. Too much data is presented in Table 1 for the reader to absorb and also much of this information is not relevant to the present study. I would recommend deleting most of the laboratory data values.

Laboratory data values have been deleted in response to this comment.

  1. There are conflicting findings in the literature regarding outcomes in patients medicated vs. not medicated for depression, over and above the issue of adverse effects of tricyclic medications. For example, There is conflicting information as to whether taking antidepressants are associated with adverse outcomes in patients with CHD. For example, one analysis from the ENRICHD study of post-MI patients (Taylor et al. Arch Gen Psychiatry) showed better outcomes in those taking SSRI's, whereas the present study and others find adverse effects of antidepressant use in patients with suspected CAD (e,g,  Krantz et al., from the WISE study, Heart; and other studies). The authors do not review the conflicting evidence in the literature based on studies using depression medication as a proxy for depression. In addition, I may have missed it in the lengthy reference list, but the set of SADHEART intitial study as well as the SADHEART-CHF clinical trial of heart failure patients did not show adverse effects outcome in medicated patients. The authors need to consider these conflicting findings in their discussion. 

In response to the reviewer’s comments, we have expanded the discussion to include a discussion of treatment efficacy of anti-depressant medications, and further expanded our existing discussion of treatment risks.

Round 2

Reviewer 1 Report

Unfortunately, the authors did not address the main points of my review. If there are no valid diagnoses of depression and anxiety disorder in the cohort, this should not be obscured. I appreciate the additional analysis and information that 73% of the individual of the depressed group were classified as depressed based on the prescription of medication and 23% based on a diagnosis of a depressive disorder. However, concerning anxiety disorders, the only classifier was the prescription of benzodiazepines. The authors must acknowledge that many antidepressants are approved for the treatment of anxiety disorders and are often prescribed for sleep disorders. The determination of mental disorders or the differentiation of anxiety disorders from depressive disorders based on the intake of benzodiazepines and antidepressants is obsolete.

  1. The title should be changed to something like "Treatment of Heart Failure Patients with Benzodiazepines and Antidepressants is Associated with Increased Mortality".
  2. The authors should not refer to diagnoses but the prescription or intake of psychiatric drugs.
  3. The authors should discuss the high rate of prescription of antidepressants, although the current evidence shows that antidepressants do not have a favorable effect in HF (but are associated with premature death).
  4. The authors should discuss the lack of diagnostic awareness. This indicates the stigmatization of mentally ill patients and corresponding deficits in medical education.
  5. The majority of case determinations were based on the intake of medication. The small minority of those where this was based on a clinical diagnosis and not on medication should be excluded from the principal analysis.
  6. Important studies are still missing, to which the authors must refer in the introduction or discussion.

References
Angermann, C.E., Gelbrich, G., Stork, S., Gunold, H., Edelmann, F., Wachter, R., Schunkert, H., Graf, T., Kindermann, I., Haass, M., Blankenberg, S., Pankuweit, S., Prettin, C., Gottwik, M., Bohm, M., Faller, H., Deckert, J., Ertl, G., 2016. Effect of Escitalopram on All-Cause Mortality and Hospitalization in Patients With Heart Failure and Depression: The MOOD-HF Randomized Clinical Trial. Jama 315, 2683-2693.
Biffi, A., Rea, F., Scotti, L., Lucenteforte, E., Vannacci, A., Lombardi, N., Chinellato, A., Onder, G., Vitale, C., Cascini, S., Ingrasciotta, Y., Roberto, G., Mugelli, A., Corrao, G., 2020. Antidepressants and the Risk of Cardiovascular Events in Elderly Affected by Cardiovascular Disease: A Real-Life Investigation From Italy. Journal of clinical psychopharmacology 40, 112-121.
Das, A., Roy, B., Schwarzer, G., Silverman, M.G., Ziegler, O., Bandyopadhyay, D., Philpotts, L.L., Sinha, S., Blumenthal, J.A., Das, S., 2019. Comparison of treatment options for depression in heart failure: A network meta-analysis. J Psychiatr Res 108, 7-23.
Fraguas, R., da Silva Telles, R.M., Alves, T.C., Andrei, A.M., Rays, J., Iosifescu, D.V., Wajngarten, M., 2009. A double-blind, placebo-controlled treatment trial of citalopram for major depressive disorder in older patients with heart failure: the relevance of the placebo effect and psychological symptoms. Contemp Clin Trials 30, 205-211.
He W, Zhou Y, Ma J, Wei B, Fu Y. Effect of antidepressants on death in patients with heart failure: a systematic review and meta-analysis. Heart Fail Rev. 2020 Nov;25(6):919-926.
O'Connor, C.M., Jiang, W., Kuchibhatla, M., Silva, S.G., Cuffe, M.S., Callwood, D.D., Zakhary, B., Stough, W.G., Arias, R.M., Rivelli, S.K., Krishnan, R., 2010. Safety and efficacy of sertraline for depression in patients with heart failure: results of the SADHART-CHF (Sertraline Against Depression and Heart Disease in Chronic Heart Failure) trial. J Am Coll Cardiol 56, 692-699.

Author Response

1. The title should be changed to something like "Treatment of Heart Failure Patients with Benzodiazepines and Antidepressants is Associated with Increased Mortality".

Authors’ response: The title has been changed to “Treatment of heart failure patients  with anxiolytics is associated with adverse outcomes, with and without depression.”

2. The authors should not refer to diagnoses but the prescription or intake of psychiatric drugs.

Authors’ response: This has been changed throughout the manuscript.

3.  The authors should discuss the high rate of prescription of antidepressants, although the current evidence shows that antidepressants do not have a favorable effect in HF (but are associated with premature death).

Authors’ response: The authors added to the discussion the comment that “Of note, the frequency of filling prescriptions for anti-depressants and benzodiazepines increases as the patient approaches death, and may reflect increased discomfort as death approaches…. This occurs despite studies that suggest that therapy with SSRIs is not effective in the treatment of depression in HF patients.” 

4. The authors should discuss the lack of diagnostic awareness. This indicates the stigmatization of mentally ill patients and corresponding deficits in medical education. 

Authors’ response: The authors appreciate the sensitivity of the reviewer to issues of stigmatization of patients challenged with mental health issues, and the need for education of health care providers to address this issue.  Although we agree that this is an important consideration, we believe that this discussion is beyond the scope of this paper.

5. The majority of case determinations were based on the intake of medication. The small minority of those where this was based on a clinical diagnosis and not on medication should be excluded from the principal analysis.

Authors’ response:  We respectfully disagree with the reviewer.  27% of the 1688    patients were diagnosed with depression but were not treated with anti-depressant medications.  This represents a significant number of patients confronting depression, with the associated excess morbidity and mortality.  In order to address the reviewer’s pertinent concern, we compared survival in patients treated with anti-depressants and patients diagnosed with depression but not treated with anti-depressants, and did not find a significant difference in the survival curves (Figure 3.)

  1. Important studies are still missing, to which the authors must refer in the introduction or discussion.

Authors’ response:  All the studies cited by the reviewer in the earlier review have been included in the manuscript, and additional studies have been added. We are open to suggestions of other pertinent studies that should be included.

Reviewer 2 Report

This is a revised manuscript and, while the authors have been responsive to the prior review, a few key weaknesses remain.  Perhaps the greatest weakness concerns a failure to account for time.  Specifically, the authors collapse a years worth of data - the year for which they gather data on the prescribing of anxiolytic/antidepressant medication.  This is treated as a 'yes/no' for the entire year, and then events subsequent to that year are treated as the dependent variable.  A person prescribed one of these medications in January of 2017 is treated the same way as a person prescribed these medications in December of 2017, yet the first person has an additional 12 months of exposure - to the medication and/or the associated diagnosis.  Furthermore, a years worth of events are excluded due to censoring.  A better approach would be to treat the independent variable as time varying, as this would address the3 key weakness described.  Absent such an approach, the results of the analyses are at best difficult to interpret, and at worst, so flawed as to render the findings almost meaningless.

A second weakness - which is acknowledged to a degree in limitations - concerns the conflating of medication prescription with diagnosis.  While I appreciate that the nature of the dataset and clinical practice makes it impossible to more accurately ascertain whether a HF patient had depression and/or anxiety, once this is stated, the authors then throughout the manuscript use anxiety and depression as descriptors, rather than using medication prescription as the descriptors.  While this problem is somewhat less for depression - where there are some diagnoses made - here too the overlap of diagnosis with prescription is not ideal.  As the authors note, some antidepressants are prescribed for anxiety, and some anxiolytics are prescribed for sleep problems, which are common among HF patients.

Author Response

  1. This is a revised manuscript and, while the authors have been responsive to the prior review, a few key weaknesses remain.  Perhaps the greatest weakness concerns a failure to account for time.  Specifically, the authors collapse a years worth of data - the year for which they gather data on the prescribing of anxiolytic/antidepressant medication.  This is treated as a 'yes/no' for the entire year, and then events subsequent to that year are treated as the dependent variable.  A person prescribed one of these medications in January of 2017 is treated the same way as a person prescribed these medications in December of 2017, yet the first person has an additional 12 months of exposure - to the medication and/or the associated diagnosis.  Furthermore, a years worth of events are excluded due to censoring.  A better approach would be to treat the independent variable as time varying, as this would address the3 key weakness described.  Absent such an approach, the results of the analyses are at best difficult to interpret, and at worst, so flawed as to render the findings almost meaningless.

Authors’ response:  We acknowledge the reviewers comment. We have therefore performed additional analyses using the independent variables anxiolytic therapy and depression/anti-depression therapy as time-dependent variables and included follow-up during the entire 2 years (2017-2018) as suggested with inclusion of the entire cohort. These results are presented in Supplemental Table 2. The data are very similar to those presented in the original analysis.

  1. A second weakness - which is acknowledged to a degree in limitations - concerns the conflating of medication prescription with diagnosis.  While I appreciate that the nature of the dataset and clinical practice makes it impossible to more accurately ascertain whether a HF patient had depression and/or anxiety, once this is stated, the authors then throughout the manuscript use anxiety and depression as descriptors, rather than using medication prescription as the descriptors.  While this problem is somewhat less for depression - where there are some diagnoses made - here too the overlap of diagnosis with prescription is not ideal.  As the authors note, some antidepressants are prescribed for anxiety, and some anxiolytics are prescribed for sleep problems, which are common among HF patients.

Authors’ response:  We have revised the manuscript accordingly and anxiety has been replaced with treatment with anxiolytics throughout the manuscriptSimilarly, we have worked to clarify that the depression category includes patients with a diagnosis of depression and treatment for depression.  We acknowledge that this is limitation in the study.

Reviewer 3 Report

The authors have done a very detailed job addressing most of the comments of the reviewers within the limitations of their dataset.The manuscript is much improved as a result. I have no further comments

Author Response

 We thank the reviewer for their insightful comments, which have led to a much improved manuscript.